# Detecting Adversaries, yet Faltering to Noise?
# Leveraging Conditional Variational AutoEncoders for Adversary Detection in the Presence of Noisy Images

**Dvij Kalaria, Aritra Hazra, and Partha Pratim Chakrabarti**
Department of Computer Science and Engineering, Indian Institute of Technology Kharagpur, INDIA

## Abstract

With the rapid advancement and increased use of deep learning models in image identification, security becomes a major concern to their deployment in safety-critical systems. Since the accuracy and robustness of deep learning models are primarily attributed from the purity of the training samples, therefore the deep learning architectures are often susceptible to adversarial attacks. Adversarial attacks are often obtained by making subtle perturbations to normal images, which are mostly imperceptible to humans, but can seriously confuse the state-of-the-art machine learning models. What is so special in the slightest intelligent perturbations or noise additions over normal images that it leads to catastrophic classifications by the deep neural networks? Using statistical hypothesis testing, we find that Conditional Variational AutoEncoders (CVAE) are surprisingly good at detecting imperceptible image perturbations. In this paper, we show how CVAEs can be effectively used to detect adversarial attacks on image classification networks. We demonstrate our results over MNIST, CIFAR-10 dataset and show how our method gives comparable performance to the state-of-the-art methods in detecting adversaries while not getting confused with noisy images, where most of the existing methods falter.

*Index Terms*—**Deep Neural Networks, Adversarial Attacks, Image Classification, Variational Autoencoders, Noisy Images**

## Introduction

The phenomenal success of deep learning models in image identification and object detection has led to its wider adoption in diverse domains ranging from safety-critical systems, such as automotive and avionics (Rao and Frtunikj 2018) to healthcare like medical imaging, robot-assisted surgery, genomics etc. (Esteva et al. 2019), to robotics and image forensics (Yang et al. 2020), etc. The performance of these deep learning architectures are often dictated by the volume of correctly labelled data used during its training phases. Recent works (Szegedy et al. 2013) (Goodfellow, Shlens, and Szegedy 2014) have shown that small and carefully chosen modifications (often in terms of noise) to the input data of a neural network classifier can cause the

model to give incorrect labels. This weakness of neural networks allows the possibility of making adversarial attacks on the input image by creating perturbations which are imperceptible to humans but however are able to convince the neural network in getting completely wrong results that too with very high confidence. Due to this, adversarial attacks may pose a serious threat to deploying deep learning models in real-world safety-critical applications. It is, therefore, imperative to devise efficient methods to thwart such adversarial attacks.

Many recent works have presented effective ways in which adversarial attacks can be avoided. Adversarial attacks can be classified into whitebox and blackbox attacks. White-box attacks (Akhtar and Mian 2018) assume access to the neural network weights and architecture, which are used for classification, and thereby specifically targeted to fool the neural network. Hence, they are more accurate than blackbox attacks (Akhtar and Mian 2018) which do not assume access the model parameters. Methods for detection of adversarial attacks can be broadly categorized as – (i) statistical methods, (ii) network based methods, and (iii) distribution based methods. Statistical methods (Hendrycks and Gimpel 2016) (Li and Li 2017) focus on exploiting certain characteristics of the input images or the final logistic-unit layer of the classifier network and try to identify adversaries through their statistical inference. A certain drawback of such methods as pointed by (Carlini and Wagner 2017) is that the statistics derived may be dataset specific and same techniques are not generalized across other datasets and also fail against strong attacks like CW-attack. Network based methods (Metzen et al. 2017) (Gong, Wang, and Ku 2017) aim at specifically training a binary classification neural network to identify the adversaries. These methods are restricted since they do not generalize well across unknown attacks on which these networks are not trained, also they are sensitive to change with the amount of perturbation values such that a small increase in these values makes this attacks unsuccessful. Also, potential whitebox attacks can be designed as shown by (Carlini and Wagner 2017) which fool both the detection network as well as the adversary classifier networks. Distribution based methods (Feinman et al. 2017) (Gao et al. 2021) (Song et al. 2017) (Xu, Evans, and Qi 2017) (Jha et al. 2018) aim at finding the probability distribution

from the clean examples and try to find the probability of the input example to quantify how much they fall within the same distribution. However, some of the methods do not guarantee robust separation of randomly perturbed and adversarial perturbed images. Hence there is a high chance that all these methods tend to get confused with random noises in the image, treating them as adversaries.

To overcome this drawback so that the learned models are robust with respect to both adversarial perturbations as well as sensitivity to random noises, we propose the use of Conditional Variational AutoEncoder (CVAE) trained over a clean image set. At the time of inference, we empirically establish that the input example falls within a low probability region of the clean examples of the predicted class from the target classifier network. It is important to note here that, this method uses both the input image as well as the predicted class to detect whether the input is an adversary as opposed to some distribution based methods which use only the distribution from the input images. On the contrary, random perturbations activate the target classifier network in such a way that the predicted output class matches with the actual class of the input image and hence it falls within the high probability region. Thus, it is empirically shown that our method does not confuse random noise with adversarial noises. Moreover, we show how our method is robust towards special attacks which have access to both the network weights of CVAE as well as the target classifier networks where many network based methods falter. Further, we show that to eventually fool our method, we may need larger perturbations which becomes visually perceptible to the human eye. The experimental results shown over MNIST and CIFAR-10 datasets present the working of our proposal. In particular, the primary contributions made by our work is as follows.

(a) We propose a framework based on CVAE to detect the possibility of adversarial attacks.
(b) We leverage distribution based methods to effectively differentiate between randomly perturbed and adversarially perturbed images.
(c) We devise techniques to robustly detect specially targeted BIM-attacks (Metzen et al. 2017) using our proposed framework.

To the best of our knowledge, this is the first work which leverages use of Variational AutoEncoder architecture for detecting adversaries as well as aptly differentiates noise from adversaries to effectively safeguard learned models against adversarial attacks.

## Adversarial Attack Models and Methods

For a test example $X$, an attacking method tries to find a perturbation, $\Delta X$ such that $|\Delta X|_k \leq \epsilon_{atk}$ where $\epsilon_{atk}$ is the perturbation threshold and $k$ is the appropriate order, generally selected as 2 or $\infty$ so that the newly formed perturbed image, $X_{adv} = X + \Delta X$. Here, each pixel in the image is represented by the $\langle \text{R}, \text{G}, \text{B} \rangle$ tuple, where $\text{R}, \text{G}, \text{B} \in [0, 1]$. In this paper, we consider only white-box attacks, i.e. the attack methods which have access to the weights of the target classifier model. However, we believe that our

method should work much better for black-box attacks as they need more perturbation to attack and hence should be more easily detected by our framework. For generating the attacks, we use the library by (Li et al. 2020).

### Random Perturbation (RANDOM)
Random perturbations are simply unbiased random values added to each pixel ranging in between $-\epsilon_{atk}$ to $\epsilon_{atk}$. Formally, the randomly perturbed image is given by,

$$X_{rand} = X + \mathcal{U}(-\epsilon_{atk}, \epsilon_{atk}) \tag{1}$$

where, $\mathcal{U}(a, b)$ denote a continuous uniform distribution in the range $[a, b]$.

### Fast Gradient Sign Method (FGSM)
Earlier works by (Goodfellow, Shlens, and Szegedy 2014) introduced the generation of malicious biased perturbations at each pixel of the input image in the direction of the loss gradient $\Delta_X L(X, y)$, where $L(X, y)$ is the loss function with which the target classifier model was trained. Formally, the adversarial examples with with $l_\infty$ norm for $\epsilon_{atk}$ are computed by,

$$X_{adv} = X + \epsilon_{atk}.sign(\Delta_X L(X, y)) \tag{2}$$

FGSM perturbations with $l_2$ norm on attack bound are calculated as,

$$X_{adv} = X + \epsilon_{atk}.\frac{\Delta_X L(X, y)}{|\Delta_X L(X, y)|_2} \tag{3}$$

### Projected Gradient Descent (PGD)
Earlier works by (Kurakin, Goodfellow, and Bengio 2017) propose a simple variant of the FGSM method by applying it multiple times with a rather smaller step size than $\epsilon_{atk}$. However, as we need the overall perturbation after all the iterations to be within $\epsilon_{atk}$-ball of $X$, we clip the modified $X$ at each step within the $\epsilon_{atk}$ ball with $l_\infty$ norm.

$$X_{adv,0} = X, \tag{4a}$$

$$X_{adv,n+1} = \text{Clip}_X^{\epsilon_{atk}}\left\{ X_{adv,n} + \alpha.sign(\Delta_X L(X_{adv,n}, y)) \right\} \tag{4b}$$

Given $\alpha$, we take the no of iterations, $n$ to be $\lfloor \frac{2\epsilon_{atk}}{\alpha} + 2 \rfloor$. This attacking method has also been named as Basic Iterative Method (BIM) in some works.

### Carlini-Wagner (CW) Method
(Carlini and Wagner 2017) proposed a more sophisticated way of generating adversarial examples by solving an optimization objective as shown in Equation 5. Value of $c$ is chosen by an efficient binary search. We use the same parameters as set in (Li et al. 2020) to make the attack.

$$X_{adv} = \text{Clip}_X^{\epsilon_{atk}}\left\{ \min_\epsilon \|\epsilon\|_2 + c.f(x + \epsilon) \right\} \tag{5}$$

### DeepFool method
DeepFool (Moosavi-Dezfooli, Fawzi, and Frossard 2016) is an even more sophisticated and efficient way of generating adversaries. It works by making the perturbation iteratively towards the decision boundary so as to achieve the adversary with minimum perturbation. We use the default parameters set in (Li et al. 2020) to make the attack.

## Proposed Framework Leveraging CVAE

In this section, we present how Conditional Variational AutoEncoders (CVAE), trained over a dataset of clean images, are capable of comprehending the inherent differentiable attributes between adversaries and noisy data and separate out both using their probability distribution map.

### Conditional Variational AutoEncoders (CVAE)

Variational AutoEncoder is a type of a Generative Adversarial Network (GAN) having two components as Encoder and Decoder. The input is first passed through an encoder to get the latent vector for the image. The latent vector is passed through the decoder to get the reconstructed input of the same size as the image. The encoder and decoder layers are trained with the objectives to get the reconstructed image as close to the input image as possible thus forcing to preserve most of the features of the input image in the latent vector to learn a compact representation of the image. The second objective is to get the distribution of the latent vectors for all the images close to the desired distribution. Hence, after the variational autoencoder is fully trained, decoder layer can be used to generate examples from randomly sampled latent vectors from the desired distribution with which the encoder and decoder layers were trained.

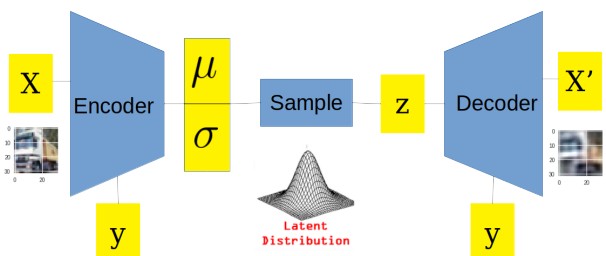

Fig. 1: CVAE Model Architecture

Conditional VAE is a variation of VAE in which along with the input image, the class of the image is also passed with the input at the encoder layer and also with the latent vector before the decoder layer (refer to Figure 1). This helps Conditional VAE to generate specific examples of a class. The loss function for CVAE is defined by Equation 6. The first term is the reconstruction loss which signifies how closely can the input $X$ be reconstructed given the latent vector $z$ and the output class from the target classifier network as condition, $c$. The second term of the loss function is the KL-divergence ($\mathcal{D}_{KL}$) between the desired distribution, $P(z|c)$ and the current distribution ($Q(z|X,c)$) of $z$ given input image $X$ and the condition $c$.

$$L(X,c) = \mathbb{E}\big[\log P(X|z,c)\big] - \mathcal{D}_{KL}\big[Q(z|X,c) \;||\; P(z|c)\big] \tag{6}$$

### Training CVAE Models

For modeling $\log P(X|z,c)$, we use the decoder neural network to output the reconstructed image, $X_{rcn}$ where we utilize the condition $c$ which is the output class of the image to get the set of parameters, $\theta(c)$ for the neural network. We calculate Binary Cross Entropy (BCE) loss of the reconstructed image, $X_{rcn}$ with the input image, $X$ to

model $\log P(X|z,c)$. Similarly, we model $Q(z|X,c)$ with the encoder neural network which takes as input image $X$ and utilizes condition, $c$ to select model parameters, $\theta(c)$ and outputs mean, $\mu$ and log of variance, $\log \sigma^2$ as parameters assuming Gaussian distribution for the conditional distribution. We set the target distribution $P(z|c)$ as unit Gaussian distribution with mean 0 and variance 1 as $N(0,1)$. The resultant loss function would be as follows,

$$
\begin{aligned}
L(X,c) \quad = \quad & \texttt{BCE}\big[X, Decoder(x \sim \mathcal{N}(\mu, \sigma^2), \theta(c))\big] \\
& -\frac{1}{2}\Big[Encoder_\sigma^2(X, \theta(c)) + Encoder_\mu^2(X, \theta(c)) \\
& \quad -1 - \log\big(Encoder_\sigma^2(X, \theta(c)))\big]
\end{aligned} \tag{7}
$$

The model architecture weights, $\theta(c)$ are a function of the condition, $c$. Hence, we learn separate weights for encoder and decoder layers of CVAE for all the classes. It implies learning different encoder and decoder for each individual class. The layers sizes are tabulated in Table I. We train the Encoder and Decoder layers of CVAE on clean images with their ground truth labels and use the condition as the predicted class from the target classifier network during inference.

| Attribute | Layer | Size |
|---|---|---|
| | Conv2d | Channels: (c, 32) |
| | | Kernel: (4,4,stride=2,padding=1) |
| | BatchNorm2d | 32 |
| | Relu | |
| | Conv2d | Channels: (32, 64) |
| Encoder | | Kernel: (4,4,stride=2,padding=1) |
| | BatchNorm2d | 64 |
| | Relu | |
| | Conv2d | Channels: (64, 128) |
| | | Kernel: (4,4,stride=2,padding=1) |
| | BatchNorm2d | 128 |
| Mean | Linear | (1024, $z_{dim}$=128) |
| Variance | Linear | (1024, $z_{dim}$=128) |
| Project | Linear | ($z_{dim}$=128, 1024) |
| | Reshape | (128,4,4) |
| | ConvTranspose2d | Channels: (128, 64) |
| | | Kernel: (4,4,stride=2,padding=1) |
| | BatchNorm2d | 64 |
| | Relu | |
| | ConvTranspose2d | Channels: (64, 32) |
| Decoder | | Kernel: (4,4,stride=2,padding=1) |
| | BatchNorm2d | 64 |
| | Relu | |
| | ConvTranspose2d | Channels: (32, c) |
| | | Kernel: (4,4,stride=2,padding=1) |
| | Sigmoid | |

TABLE I: CVAE Architecture Layer Sizes. $c$ = Number of Channels in the Input Image ($c = 3$ for CIIFAR-10 and $c = 1$ for MNIST).

### Determining Reconstruction Errors

Let $X$ be the input image and $y_{pred}$ be the predicted class obtained from the target classifier network. $X_{rcn,y_{pred}}$ is the reconstructed image obtained from the trained encoder and decoder networks with the condition $y_{pred}$. We define

the reconstruction error or the reconstruction distance as in Equation 8. The network architectures for encoder and decoder layers are given in Figure 1.

$$\texttt{Recon}(X, y) = (X - X_{rcn,y})^2 \qquad (8)$$

Two pertinent points to note here are:

- For clean test examples, the reconstruction error is bound to be less since the CVAE is trained on clean train images. As the classifier gives correct class for the clean examples, the reconstruction error with the correct class of the image as input is less.
- For the adversarial examples, as they fool the classifier network, passing the malicious output class, $y_{pred}$ of the classifier network to the CVAE with the slightly perturbed input image, the reconstructed image tries to be closer to the input with class $y_{pred}$ and hence, the reconstruction error is large.

As an example, let the clean image be a cat and its slightly perturbed image fools the classifier network to believe it is a dog. Hence, the input to the CVAE will be the slightly perturbed cat image with the class dog. Now as the encoder and decoder layers are trained to output a dog image if the class inputted is dog, the reconstructed image will try to resemble a dog but since the input is a cat image, there will be large reconstruction error. Hence, we use reconstruction error as a measure to determine if the input image is adversarial. We first train the Conditional Variational AutoEncoder (CVAE) on clean images with the ground truth class as the condition. Examples of reconstructions for clean and adversarial examples are given in Figure 2 and Figure 3.

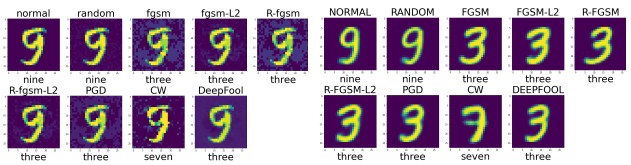

(a) Input Images    (b) Reconstructed Images

Fig. 2: Clean and Adversarial Attacked Images to CVAE from MNIST Dataset

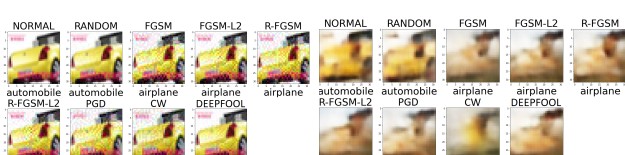

(a) Input Images    (b) Reconstructed Images

Fig. 3: Clean and Adversarial Attacked Images to CVAE from CIFAR-10 Dataset.

### Obtaining $p$-value

As already discussed, the reconstruction error is used as a basis for detection of adversaries. We first obtain the reconstruction distances for the train dataset of clean images which is expected to be similar to that of the train images. On the other hand, for the adversarial examples, as the predicted class $y$ is incorrect, the reconstruction is

expected to be worse as it will be more similar to the image of class $y$ as the decoder network is trained to generate such images. Also for random images, as they do not mostly fool the classifier network, the predicted class, $y$ is expected to be correct, hence reconstruction distance is expected to be less. Besides qualitative analysis, for the quantitative measure, we use the permutation test from (Efron and Tibshirani 1993). We can provide an uncertainty value for each input about whether it comes from the training distribution. Specifically, let the input $X'$ and training images $X_1, X_2, \ldots, X_N$. We first compute the reconstruction distances denoted by $\texttt{Recon}(X, y)$ for all samples with the condition as the predicted class $y = \texttt{Classifier}(X)$. Then, using the rank of $\texttt{Recon}(X', y')$ in $\{\texttt{Recon}(X_1, y_1), \texttt{Recon}(X_2, y_2), \ldots, \texttt{Recon}(X_N, y_N)\}$ as our test statistic, we get,

$$
\begin{aligned}
T &= T(X'; X_1, X_2, \ldots, X_N) \\
&= \sum_{i=1}^{N} I\big[\texttt{Recon}(X_i, y_i) \leq \texttt{Recon}(X', y')\big] \quad (9)
\end{aligned}
$$

Where $I[.]$ is an indicator function which returns 1 if the condition inside brackets is true, and 0 if false. By permutation principle, $p$-value for each sample will be,

$$p = \frac{1}{N+1}\Big(\sum_{i=1}^{N} I[T_i \leq T] + 1\Big) \qquad (10)$$

Larger $p$-value implies that the sample is more probable to be a clean example. Let $t$ be the threshold on the obtained $p$-value for the sample, hence if $p_{X,y} < t$, the sample $X$ is classified as an adversary. Algorithm 1 presents the overall resulting procedure combining all above mentioned stages.

---

**Algorithm 1** Adversarial Detection Algorithm

---

1: **function** DETECT_ADVERSARIES $(X_{train}, Y_{train}, X, t)$
2:   recon $\leftarrow$ Train$(X_{train}, Y_{train})$
3:   recon_dists $\leftarrow$ Recon$(X_{train}, Y_{train})$
4:   Adversaries $\leftarrow \phi$
5:   **for** $x$ in $X$ **do**
6:    $y_{pred} \leftarrow$ Classifier$(x)$
7:    recon_dist_x $\leftarrow$ Recon$(x, y_{pred})$
8:    pval $\leftarrow$ $p$-value$(recon\_dist\_x, recon\_dists)$
9:    **if** pval $\leq t$ **then**
10:     Adversaries.insert$(x)$
11:   **return** Adversaries

---

Algorithm 1 first trains the CVAE network with clean training samples (Line 2) and formulates the reconstruction distances (Line 3). Then, for each of the test samples which may contain clean, randomly perturbed as well as adversarial examples, first the output predicted class is obtained using a target classifier network, followed by finding it's reconstructed image from CVAE, and finally by obtaining it's $p$-value to be used for thresholding (Lines 5-8). Images with $p$-value less than given threshold ($t$) are classified as adversaries (Lines 9-10).

## Experimental Results

We experimented our proposed methodology over MNIST and CIFAR-10 datasets. All the experiments are performed

in Google Colab GPU having 0.82GHz frequency, 12GB RAM and dual-core CPU having 2.3GHz frequency, 12GB RAM. An exploratory version of the code-base will be made public on github.

## Datasets and Models

Two datasets are used for the experiments in this paper, namely MNIST (LeCun, Cortes, and Burges 2010) and CIFAR-10 (Krizhevsky 2009). MNIST dataset consists of hand-written images of numbers from 0 to 9. It consists of 60,000 training examples and 10,000 test examples where each image is a $28 \times 28$ gray-scale image associated with a label from 1 of the 10 classes. CIFAR-10 is broadly used for comparison of image classification tasks. It also consists of 60,000 image of which 50,000 are used for training and the rest 10,000 are used for testing. Each image is a $32 \times 32$ coloured image i.e. consisting of 3 channels associated with a label indicating 1 out of 10 classes.

We use state-of-the-art deep neural network image classifier, ResNet18 (He et al. 2015) as the target network for the experiments. We use the pre-trained model weights available from (Idelbayev ) for both MNIST as well as CIFAR-10 datasets.

## Performance over Grey-box attacks

If the attacker has the access only to the model parameters of the target classifier model and no information about the detector method or it's model parameters, then we call such attack setting as Grey-box. This is the most common attack setting used in previous works against which we evaluate the most common attacks with standard epsilon setting as used in other works for both the datasets. For MNIST, the value of $\epsilon$ is commonly used between 0.15-0.3 for FGSM attack and 0.1 for iterative attacks (Samangouei, Kabkab, and Chellappa 2018) (Gong, Wang, and Ku 2017) (Xu, Evans, and Qi 2017). While for CIFAR10, the value of $\epsilon$ is most commonly chosen to be $\frac{8}{255}$ as in (Song et al. 2017) (Xu, Evans, and Qi 2017) (Fidel, Bitton, and Shabtai 2020). For DeepFool (Moosavi-Dezfooli, Fawzi, and Frossard 2016) and Carlini Wagner (CW) (Carlini and Wagner 2017) attacks, the $\epsilon$ bound is not present. The standard parameters as used by default in (Li et al. 2020) have been used for these 2 attacks. For $L_2$ attacks, the $\epsilon$ bound is chosen such that success of the attack is similar to their $L_\infty$ counterparts as the values used are very different in previous works.

**Reconstruction Error Distribution:** The histogram distribution of reconstruction errors for MNIST and CIFAR-10 datasets for different attacks are given in Figure 4. For adversarial attacked examples, only examples which fool the network are included in the distribution for fair comparison. It may be noted that, the reconstruction errors for adversarial examples is higher than normal examples as expected. Also, reconstructions errors for randomly perturbed test samples are similar to those of normal examples but slightly larger as expected due to reconstruction error contributed from noise.

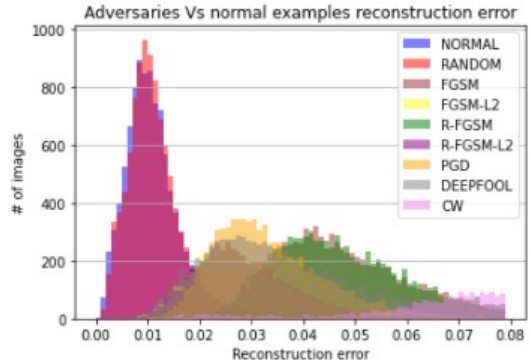

(a) MNIST dataset

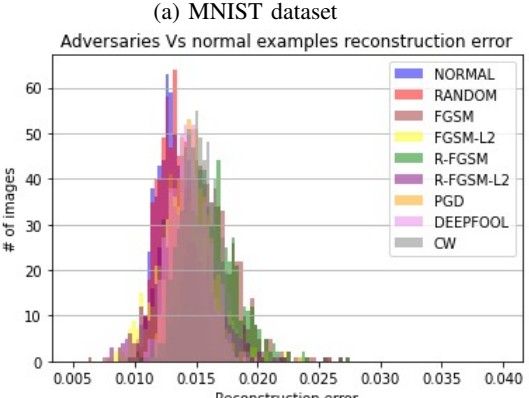

(b) CIFAR-10 dataset

Fig. 4: Reconstruction Distances for different Grey-box attacks

$p$-**value Distribution:** From the reconstruction error values, the distribution histogram of p-values of test samples for MNIST and CIFAR-10 datasets are given in Figure 5. It may be noted that, in case of adversaries, most samples have $p$-value close to 0 due to their high reconstruction error; whereas for the normal and randomly perturbed images, $p$-value is nearly uniformly distributed as expected.

**ROC Characteristics:** Using the $p$-values, ROC curves can be plotted as shown in Figure 6. As can be observed from ROC curves, clean and randomly perturbed attacks can be very well classified from all adversarial attacks. The values of $\epsilon_{atk}$ were used such that the attack is able to fool the target detector for at-least $45\%$ samples. The percentage of samples on which the attack was successful for each attack is shown in Table II.

**Statistical Results and Discussions:** The statistics for clean, randomly perturbed and adversarial attacked images for MNIST and CIFAR datasets are given in Table II. Error rate signifies the ratio of the number of examples which were misclassified by the target network. Last column (AUC) lists the area under the ROC curve. The area for adversaries is expected to be close to 1; whereas for the normal and randomly perturbed images, it is expected to be around $0.5$.

It is worthy to note that, the obtained statistics are much comparable with the state-of-the-art results as tabulated in

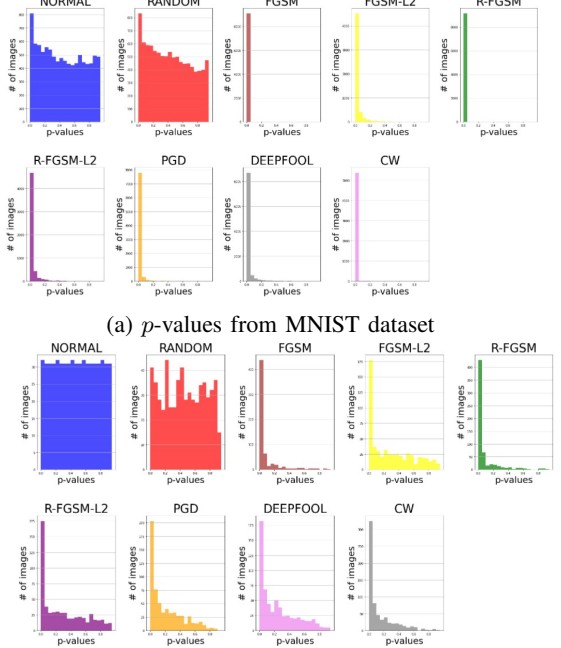

(a) *p*-values from MNIST dataset

(b) *p*-values from CIFAR-10 dataset

Fig. 5: Generated *p*-values for different Grey-box attacks

| Type | Error Rate (%) | | Parameters | | AUC | |
|---|---|---|---|---|---|---|
| | MNIST | CIFAR-10 | MNIST | CIFAR-10 | MNIST | CIFAR-10 |
| NORMAL | 2.2 | 8.92 | - | - | 0.5 | 0.5 |
| RANDOM | 2.3 | 9.41 | $\epsilon$=0.1 | $\epsilon=\frac{8}{255}$ | 0.52 | 0.514 |
| FGSM | 90.8 | 40.02 | $\epsilon$=0.15 | $\epsilon=\frac{8}{255}$ | 0.99 | 0.91 |
| FGSM-L2 | 53.3 | 34.20 | $\epsilon$=1.5 | $\epsilon=1$ | 0.95 | 0.63 |
| R-FGSM | 91.3 | 41.29 | $\epsilon$=(0.05,0.1) | $\epsilon=(\frac{4}{255},\frac{8}{255})$ | 0.99 | 0.91 |
| R-FGSM-L2 | 54.84 | 34.72 | $\epsilon$=(0.05,1.5) | $\epsilon=(\frac{4}{255},1)$ | 0.95 | 0.64 |
| PGD | 82.13 | 99.17 | $\epsilon$=0.1,n=12 $\epsilon_{step}=0.02$ | $\epsilon=\frac{8}{255}$,n=12 $\epsilon_{step}=\frac{1}{255}$ | 0.974 | 0.78 |
| CW | 100 | 100 | - | - | 0.98 | 0.86 |
| DeepFool | 97.3 | 93.89 | - | - | 0.962 | 0.75 |

TABLE II: Image Statistics for MNIST and CIFAR-10. AUC : Area Under the ROC Curve. Error Rate (%) : Percentage of samples mis-classified or Successfully-attacked

Table V (Given in the **Appendix**). Interestingly, some of the methods (Song et al. 2017) explicitly report comparison results with randomly perturbed images and are ineffective in distinguishing adversaries from random noises, but most other methods do not report results with random noise added to the input image. Since other methods use varied experimental setting, attack models, different datasets as well as $\epsilon_{atk}$ values and network model, exact comparisons with other methods is not directly relevant primarily due to such varied experimental settings. However, the results reported within the Table V (Given in the **Appendix**) are mostly similar to our results while our method is able to statistically differentiate from random noisy images.

In addition to this, since our method does not use any adversarial examples for training, it is not prone to changes in value of $\epsilon$ or with change in attacks which network based methods face as they are explicitly trained with known values of $\epsilon$ and types of attacks. Moreover, among

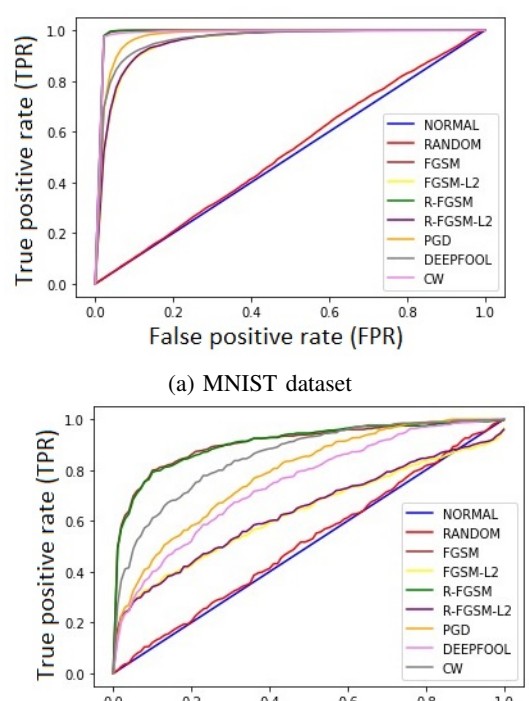

(a) MNIST dataset

(b) CIFAR-10 dataset

Fig. 6: ROC Curves for different Grey-box attacks

distribution and statistics based methods, to the best of our knowledge, utilization of the predicted class from target network has not been done before. Most of these methods either use the input image itself (Jha et al. 2018) (Song et al. 2017) (Xu, Evans, and Qi 2017), or the final logits layer (Feinman et al. 2017) (Hendrycks and Gimpel 2016), or some intermediate layer (Li and Li 2017) (Fidel, Bitton, and Shabtai 2020) from target architecture for inference, while we use the input image and the predicted class from target network to do the same.

**Performance over White-box attacks**

In this case, we evaluate the attacks if the attacker has the information of both the defense method as well as the target classifier network. (Metzen et al. 2017) proposed a modified PGD method which uses the gradient of the loss function of the detector network assuming that it is differentiable along with the loss function of the target classifier network to generate the adversarial examples. If the attacker also has access to the model weights of the detector CVAE network, an attack can be devised to fool both the detector as well as the classifier network. The modified PGD can be expressed as follows :-

$$X_{adv,0} = X, \tag{11a}$$

$$X_{adv,n+1} = \mathtt{Clip}_X^{\epsilon_{atk}} \Big\{ X_{adv,n} + $$
$$\alpha.sign\big( (1-\sigma).\Delta_X L_{cls}(X_{adv,n}, y_{target}) + $$
$$\sigma.\Delta_X L_{det}(X_{adv,n}, y_{target}) \big) \Big\} \tag{11b}$$

Where $y_{target}$ is the target class and $L_{det}$ is the reconstruction distance from Equation 8. It is worthy to note that our proposed detector CVAE is differentiable only for the targeted attack setting. For the non-targeted attack, as the condition required for the CVAE is obtained from the target classifier output which is discrete, the differentiation operation is not valid. We set the target randomly as any class other than the true class for testing.

**Effect of $\sigma$:** To observe the effect of changing value of $\sigma$, we keep the value of $\epsilon$ fixed at 0.1. As can be observed in Figure 7, the increase in value of $\sigma$ implies larger weight on fooling the detector i.e. getting less reconstruction distance. Hence, as expected the attack becomes less successful with larger values of $\sigma$ 8 and gets lesser AUC values 7, hence more effectively fooling the detector. For CIFAR-10 dataset, the detection model does get fooled for higher $c$-values but however the error rate is significantly low for those values, implying that only a few samples get attacks on setting such value.

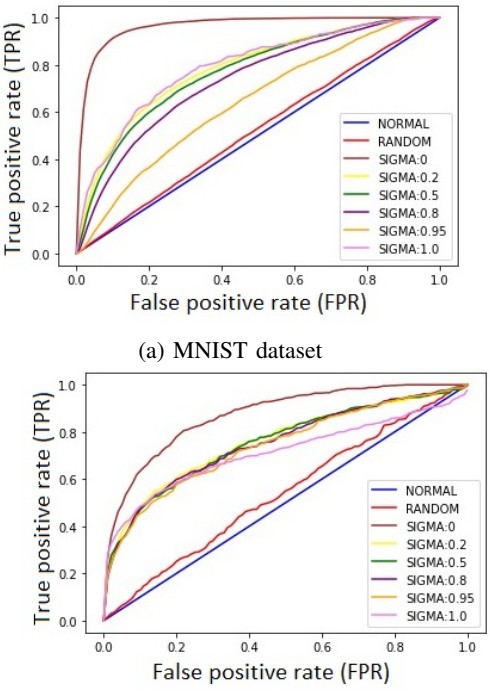

(a) MNIST dataset

(b) CIFAR-10 dataset

Fig. 7: ROC Curves for different values of $\sigma$. More area under the curve implies better detectivity for that attack. With more $\sigma$ value, the attack, as the focus shifts to fooling the detector, it becomes difficult for the detector to detect.

**Effect of $\epsilon$:** With changing values of $\epsilon$, there is more space available for the attack to act, hence the attack becomes more successful as more no of images are attacked as observed in Figure 10. At the same time, the trend for AUC curves is shown in Figure 9. The initial dip in the value is as expected as the detector tends to be fooled with larger $\epsilon$ bound. From both these trends, it can be noted that

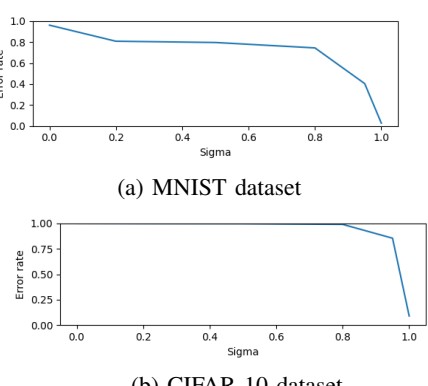

(a) MNIST dataset

(b) CIFAR-10 dataset

Fig. 8: Success rate for different values of $\sigma$. More value of $\sigma$ means more focus on fooling the detector, hence success rate of fooling the detector decreases with increasing $\sigma$.

for robustly attacking both the detector and target classifier for significantly higher no of images require significantly larger attack to be made for both the datasets.

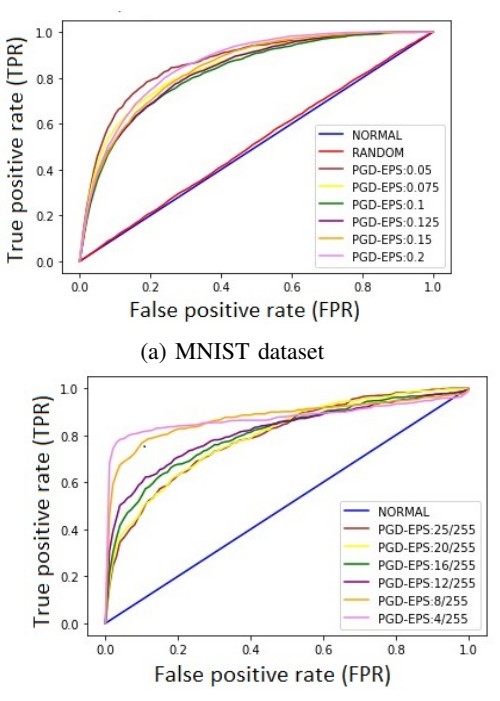

(a) MNIST dataset

(b) CIFAR-10 dataset

Fig. 9: ROC Curves for different values of $\epsilon$. With more $\epsilon$ value, due to more space available for the attack, attack becomes less detectable on average.

## Related Works

There has been an active research in the direction of adversaries and the ways to avoid them, primarily these methods are statistical as well as machine learning (neural network) based which produces systematic identification and rectification of images into desired target classes.

**Statistical Methods:** Statistical methods focus on exploiting certain characteristics of the input images and try

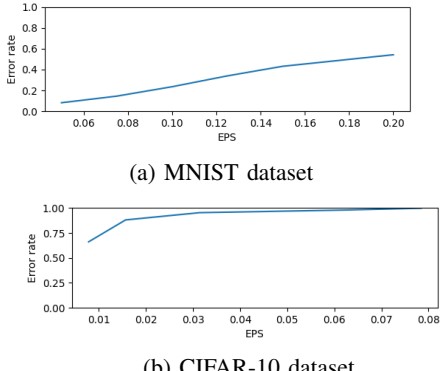

(a) MNIST dataset

(b) CIFAR-10 dataset

Fig. 10: Success rate for different values of $\epsilon$. More value of $\epsilon$ means more space available for the attack, hence success rate increases

to identify adversaries through their statistical inference. Some early works include use of PCA, softmax distribution of final layer logits (Hendrycks and Gimpel 2016), reconstruction from logits (Li and Li 2017) to identify adversaries. Carlini and Wagner (Carlini and Wagner 2017) showed how these methods are not robust against strong attacks and most of the methods work on some specific datasets but do not generalize on others as the same statistical thresholds do not work.

**Network based Methods:** Network based methods aim at specifically training a neural network to identify the adversaries. Binary classification networks (Metzen et al. 2017) (Gong, Wang, and Ku 2017) are trained to output a confidence score on the presence of adversaries. Some methods propose addition of a separate classification node in the target network itself (Hosseini et al. 2017). The training is done in the same way with the augmented dataset. (Carrara et al. 2018) uses feature distant spaces of intermediate layer values in the target network to train an LSTM network for classifying adversaries. Major challenges faced by these methods is that the classification networks are differentiable, thus if the attacker has access to the weights of the model, a specifically targeted attack can be devised as suggested by Carlini and Wagner (Carlini and Wagner 2017) to fool both the target network as well as the adversary classifier. Moreover, these methods are highly sensitive to the perturbation threshold set for adversarial attack and fail to identify attacks beyond a preset threshold.

**Distribution based Methods:** Distribution based methods aim at finding the probability distribution from the clean examples and try to find the probability of the input example to fall within the same distribution. Some of these methods include using Kernel Density Estimate on the logits from the final softmax layer (Feinman et al. 2017). (Gao et al. 2021) used Maximum mean discrepancy (MMD) from the distribution of the input examples to classify adversaries based on their probability of occurrence in the input distribution. PixelDefend (Song et al. 2017) uses PixelCNN to get the Bits Per Dimension (BPD) score for the input image. (Xu, Evans, and Qi 2017) uses the difference in the final logit vector for original and squeezed

images as a medium to create distribution and use it for inference. (Jha et al. 2018) compares different dimensionality reduction techniques to get low level representations of input images and use it for bayesian inference to detect adversaries.

Some other special methods include use of SHAP signatures (Fidel, Bitton, and Shabtai 2020) which are used for getting explanations on where the classifier network is focusing as an input for detecting adversaries.

*A detailed comparative study with all these existing approaches is summarized through Table V in the* **Appendix**.

## Comparison with State-of-the-Art using Generative Networks

Finally we compare our work with these 3 works (Meng and Chen 2017) (Hwang et al. 2019) (Samangouei, Kabkab, and Chellappa 2018) proposed earlier which uses Generative networks for detection and purification of adversaries. We make our comparison on MNIST dataset which is used commonly in the 3 works (Table III). Our results are typically the best for all attacks or are off by short margin from the best. For the strongest attack, our performance is much better. This show how our method is more effective while not being confused with random perturbation as an adversary. More details are given in the **Appendix**.

| Type | AUC | | | |
|---|---|---|---|---|
| | MagNet | PuVAE | DefenseGAN | CVAE (Ours) |
| RANDOM | 0.61 | 0.72 | 0.52 | **0.52** |
| FGSM | 0.98 | 0.96 | 0.77 | **0.99** |
| FGSM-L2 | 0.84 | 0.60 | 0.60 | **0.95** |
| R-FGSM | **0.989** | 0.97 | 0.78 | 0.987 |
| R-FGSM-L2 | 0.86 | 0.61 | 0.62 | **0.95** |
| PGD | **0.98** | 0.95 | 0.65 | 0.97 |
| CW | 0.983 | 0.92 | 0.94 | **0.986** |
| DeepFool | 0.86 | 0.86 | 0.92 | **0.96** |
| **Strongest** | 0.84 | 0.60 | 0.60 | **0.95** |

TABLE III: Comparison in ROC AUC statistics with other methods. More AUC implies more detectablity. 0.5 value of AUC implies no detection. For RANDOM, value close to 0.5 is better while for adversaries, higher value is better.

## Conclusion

In this work, we propose the use of Conditional Variational AutoEncoder (CVAE) for detecting adversarial attacks. We utilized statistical base methods to verify that the adversarial attacks usually lie outside of the training distribution. We demonstrate how our method can specifically differentiate between random perturbations and targeted attacks which is necessary for some applications where the raw camera image may contain random noises which should not be confused with an adversarial attack. Furthermore, we demonstrate how it takes huge targeted perturbation to fool both the detector as well as the target classifier. Our framework presents a practical, effective and robust adversary detection approach in comparison to existing state-of-the-art techniques which falter to differentiate noisy data from adversaries. As a possible future work, it would be interesting to see the use of Variational AutoEncoders for automatically purifying the adversarialy attacked images.

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

# Appendix

## Use of simple AutoEncoder (AE)

MagNet (Meng and Chen 2017) uses AutoEncoder (AE) for detecting adversaries. We compare the results with our proposed CVAE architecture on the same experiment setting and present the comparison in AUC values of the ROC curve observed for the 2 cases. Although the paper's claim is based on both detection as well as purification (if not detected) of the adversaries. MagNet uses their detection framework for detecting larger adversarial perturbations which cannot be purified. For smaller perturbations, MagNet proposes to purify the adversaries by a different AutoEncoder model. We make the relevant comparison only for the detection part with our proposed method. Using the same architecture as proposed, our results are better for the strongest attack while not getting confused with random perturbations of similar magnitude. ROC curves obtained for different adversaries for MagNet are given in Figure 11

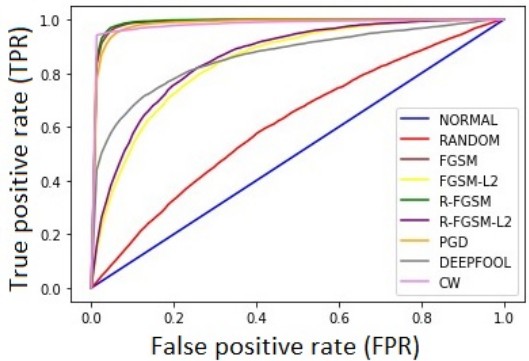

Fig. 11: ROC curve of different adversaries for MagNet

## Use of Variational AutoEncoder (VAE)

PuVAE (Hwang et al. 2019) uses Variational AutoEncoder (VAE) for purifying adversaries. We compare the results with our proposed CVAE architecture on the same experiment setting. PuVAE however, does not propose using VAE for detection of adversaries but in case if their model is to be used for detection, it would be based on the reconstruction distance. So, we make the comparison with our proposed CVAE architecture. ROC curves for different adversaries are given in Figure 12

## Use of Generative Adversarial Network (GAN)

Defense-GAN (Samangouei, Kabkab, and Chellappa 2018) uses Generative Adversarial Network (GAN) for detecting adversaries. We used $L = 100$ and $R = 10$ for getting the results as per our experiment setting. We compare the results with our proposed CVAE architecture on the same experiment setting and present the comparison in AUC values of the ROC curve observed for the 2 cases. Although the paper's main claim is about purification of the adversaries, we make the relevant comparison for the detection part with our proposed method. We used the

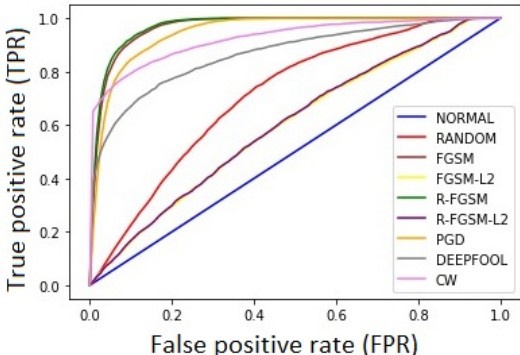

Fig. 12: ROC curve of different adversaries for PuVAE

same architecture as mentioned in (Samangouei, Kabkab, and Chellappa 2018) and got comparable results as per their claim for MNIST dataset on FGSM adversaries. As this method took a lot of time to run, we randomly chose 1000 samples out of 10000 test samples for evaluation due to time constraint. The detection performance for other attacks is considerably low. Also, Defense-GAN is quite slow as it needs to solve an optimization problem for each image to get its corresponding reconstructed image. Average computation time required by Defense-GAN is $2.8s$ per image while our method takes $0.17s$ per image with a batch size of 16. Hence, our method is roughly 16 times faster than Defense-GAN. Refer to Figure 13 for the ROC curves for Defense-GAN.

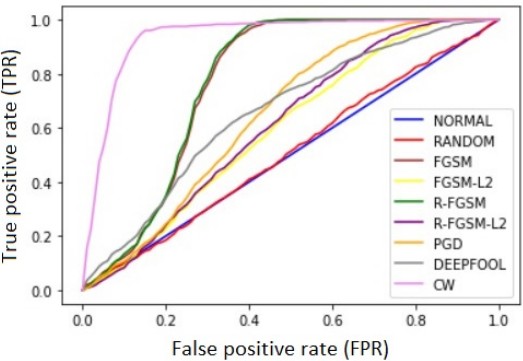

Fig. 13: ROC curve of different adversaries for DefenseGan

## Reporting the results in robust detection risk form

(Tramèr 2021) argued that most of the results reported for detection form are inconsistent and there seems to be a fair chance for works to over-claim the detection results. (Tramèr 2021) shows a reduction from robust detection for a given $\epsilon$ bound to robust purification of images within $\frac{\epsilon}{2}$ by the same margin of error. This means that a robust detector being able to detect all adversaries within $\epsilon$ bound is equivalent to a robust (but inefficient) purifier that purifies all adversaries within $\frac{\epsilon}{2}$ bound. While, using Area Under the Curve (AUC) of the full ROC curves can be a good way for comparison of different detectors, we additionally

present results in the robust detection risk form (Equation 12) as suggested by (Tramèr 2021). The upper bound on value of robust risk ($R^{\epsilon}_{adv-det}$) can be obtained by Equation 13. We choose appropriate FPR from the ROC curve such that the robust risk ($R^{\epsilon,upper}_{adv-det}$) gets minimised. The results for grey-box attacks are reported in table IV.

$$R^{\epsilon}_{adv-det} \leq FPR + FNR + E_{normal} \qquad (12)$$

$$R^{\epsilon,upper}_{adv-det} = Min_t(FPR_t + FNR_t + E_{normal}) \qquad (13)$$

| Type | Parameters | | $R^{\epsilon,upper}_{adv-det}$ | |
|---|---|---|---|---|
| | MNIST | CIFAR-10 | MNIST | CIFAR-10 |
| FGSM | $\epsilon$=0.15 | $\epsilon=\frac{8}{255}$ | 0.04 | 0.38 |
| FGSM-L2 | $\epsilon$=1.5 | $\epsilon = 1$ | 0.21 | 0.79 |
| R-FGSM | $\epsilon$=(0.05,0.1) | $\epsilon=(\frac{4}{255},\frac{8}{255})$ | 0.05 | 0.39 |
| R-FGSM-L2 | $\epsilon$=(0.05,1.5) | $\epsilon=(\frac{4}{255},1)$ | 0.22 | 0.81 |
| PGD | $\epsilon$=0.1,$n$=12 $\epsilon_{step} = 0.02$ | $\epsilon=\frac{8}{255}$,$n$=12 $\epsilon_{step}=\frac{1}{255}$ | 0.16 | 0.59 |
| CW | - | - | 0.08 | 0.47 |
| DeepFool | - | - | 0.18 | 0.61 |

TABLE IV: Robust detection statistics for MNIST and CIFAR-10. $E_{normal}$ for MNIST is 0.022 and for CIFAR-10 is 0.089

| References | Concepts Established | Datasets Used | Attack Types | Primary Results | Major Shortcomings | Advantages of our Proposed Work |
|---|---|---|---|---|---|---|
| (Hendrycks and Gimpel 2016) | PCA whitening on distribution of final softmax layer | MNIST, CIFAR-10, Tiny-ImageNet | FGSM($l_\infty$), BIM($l_\infty$) | AUC ROC for CIFAR-10: FGSM($l_\infty$) = 0.928, BIM($l_\infty$) = 0.912 | Not tested for strong attacks, Not tested to differentiate random noisy images | Ability to differentiate from randomly perturbed images, evaluation against strong attacks and target classifier. |
| (Li and Li 2017) | Cascade classifier based PCA statistics of intermediate convolution layers | ILSVRC-2012 | L-BGFS (Similar to CW) | AUC of ROC: 0.908 | Not tested for strong attacks, standard datsets, for random noises | Ability to differentiate from randomly perturbed images, evaluation against strong and wider attacks. |
| (Metzen et al. 2017) | Binary classifier network with intermediate layer features as input | CIFAR-10 | FGSM ($l_2,l_\infty$), BIM ($l_2,l_\infty$), DeepFool, Dynamic BIM (Similar to S-BIM) | Highest detection accuracy among different layers: FGSM = 0.97, BIM($l_2$) = 0.8, BIM($l_\infty$) = 0.82, DeepFool($l_2$) = 0.72, DeepFool($l_\infty$) = 0.75, Dynamic-BIM = 0.8 (Average) | Need to train with adversarial examples, hence do not generalize well on other attacks, not evaluated for random noisy images | No use of adversaries for training, ability to differentiate from randomly perturbed images, more robust to dynamic adversaries, better AUC results |
| (Gong, Wang, and Ku 2017) | Binary classifier network trained with input image | MNIST, CIFAR-10, SVHN | FGSM($l_\infty$), TGSM($L\infty$), JSMA | Average accuracy of 0.9914 (MNIST), 0.8279 (CIFAR-10), 0.9378 (SVHN) | Trained with generated adversaries, hence does not generalize well on other adversaries, sensitive to $\epsilon$ changes | No use of adversaries for training, ability to differentiate from randomly perturbed images |
| (Carrara et al. 2018) | LSTM on distant features at each layer of target classifier network | ILSVRC dataset | FGSM, BIM, PGD, L-BFGS ($L\infty$) | ROC AUC: FGSM = 0.996, BIM = 0.997, L-BFGS = 0.854, PGD = 0.997 | Not evaluated for differentiation from random noisy images, on special attack which has access to network weights | No use of adversaries for training, ability to differentiate from randomly perturbed images, evaluaion on $l_2$ attacks |
| (Feinman et al. 2017) | Bayesian density estimate on final softmax layer | MNIST, CIFAR-10, SVHN | FGSM, BIM, JSMA, CW ($l_\infty$) | CIFAR-10 ROC-AUC: FGSM = 0.9057, BIM = 0.81, JSMA = 0.92, CW = 0.92 | No explicit test for random noisy images | Ability to differentiate between randomly perturbed images, better AUC values |
| (Song et al. 2017) | Using PixelDefend to get reconstruction error on input image | Fashion MNIST, CIFAR-10 | FGSM, BIM, DeepFool, CW ($L\infty$) | ROC curves given, AUC not given | Cannot differentiate random noisy images from adversaries | Ability to differentiate between randomly perturbed and clean images |
| (Xu, Evans, and Qi 2017) | Feature squeezing and comparison | MNIST, CIFAR-10, ImageNet | FGSM, BIM, DeepFool, JSMA, CW | Overall detection rate: MNIST = 0.982, CIFAR-10 = 0.845, ImageNet = 0.859 | No test for randomly perturbed images | Ability to differentiate from randomly perturbed images, better AUC values |
| (Jha et al. 2018) | Using bayesian inference from manifolds on input image | MNIST, CIFAR-10 | FGSM, BIM | No quantitative results reported | No comparison without quantitative results | Ability to differentiate from randomly perturbed images, evaluation against strong attacks |
| (Fidel, Bitton, and Shabtai 2020) | Using SHAP signatures of input image | MNIST, CIFAR-10 | FGSM, BIM, DeepFool etc. | Average ROC-AUC: CIFAR-10 = 0.966, MNIST = 0.967 | Not tested for random noisy images | No use of adversaries for training, ability to differentiate from randomly perturbed images |

TABLE V: Summary of Related Works and Comparative Study with these Existing Methods