# OpenReview forum: "Detecting Adversaries, yet Faltering to Noise? Leveraging Conditional Variational AutoEncoders for Adversary Detection in the Presence of Noisy Images"
_AAAI.org/2022/Workshop/AdvML — AAAI-22 AdvML Workshop LongPaper_

### Official Review · Reviewer_L2gD · 2021-11-30
**Review of Paper 23**

**Rating:** 7
**Confidence:** 4

**Review:**

This paper propose to use CVAE to detect adversarial examples, while avoid be sensitive on random noisy samples. The experiments are done on MNIST and CIFAR-10, under different attacks and threat models.

The idea of exploiting the information of predicted labels is reasonable, and the authors also evaluate the method under white-box attacks (i.e., adaptive attacks). However, recent work [1] find that many detection-based defenses may over-claim their performance, so it would be more convinced if the authors can convert their reported results into classification-based defenses and do a sanity check along with [1].

[1] Detecting Adversarial Examples Is (Nearly) As Hard As Classifying Them.

---

### Official Review · Reviewer_1NMu · 2021-12-02
**Good Approach and Extensive Evaluation**

**Rating:** 7
**Confidence:** 3

**Review:**

**Summary**:

In this work, the authors have described a novel approach towards detecting adversarial perturbations in images. The approach uses conditional VAEs trained on clean (non-perturbed) images, and leverages that adversarial perturbed examples actually come from a different distribution than the predicted class and hence will have a higher reconstruction error. The authors evaluate their approach over several known black-box and white-box methods.

**PROS**:

1. The authors address the distinction between random noise and adversarial perturbation, which is important since not all random noise will be adversarial.
2. The authors extensively evaluate their approach over known attacks, which shows that the CVAE based method can be of practical significance.
3. The approach leverages correctly the fundamental characteristics of autoencoders -- that since adversarial examples will cause only imperceptible distribution shifts in the feature space, the reconstruction error is high. In other words, the tradeoff between classification label and adversarial perturbation is leveraged.

**CONS**
1. In the introduction, the authors explain drawbacks of statistical, network-based and distribution methods such as domain-dependency and non-transferability. It would be also good to address whether the CVAE approach overcomes any of these issues, and if so, how.

---

### Decision · Program_Chairs · 2021-12-02

**Decision:**

Accept (Long Paper)

**Comment:**

The reviewer agrees to accept the paper. Please consider the reviewer's comment in the final version.